# MFTSGNet: A Dual-Branch Spatio-Temporal Network for Robust Seizure Detection

Zhiheng Zhang[1#]
*School of Data Science*
*The Chinese University of Hong Kong, Shenzhen*
Shenzhen, China
123090850@link.cuhk.edu.cn

Xiang Li[2#]
*School of Medicine*
*The Chinese University of Hong Kong, Shenzhen*
Shenzhen, China
224050069@link.cuhk.edu.cn

Zhaonian Guo[3]
*School of Sicence and Engineering*
*The Chinese University of Hong Kong, Shenzhen*
Shenzhen, China
122090144@link.cuhk.edu.cn

Xin Wang[4]
*Research Center for Neural Engineering*
*Shenzhen Institutes of Advanced Technology*
Shenzhen, China
wangxin@siat.ac.cn

Ke Zhang[5]
*School of Medicine*
*The Chinese University of Hong Kong, Shenzhen*
Shenzhen, China
kezhang@cug.edu.cn

Shixiong Chen*
*School of Medicine*
*The Chinese University of Hong Kong, Shenzhen*
Shenzhen, China
chenshixiong@cuhk.edu.cn

*Abstract*—Epileptic seizure detection from EEG signals remains a challenging task due to the complex spatial-temporal dependencies in neural activities. In this paper, we propose a novel model named Multi-Frequency TCN Spatial Graph Network(MFTSGNet), which incorporates a dual-branch architecture combining Wavelet Transform with Temporal Convolution (WT+TCN) for time-frequency analysis and a Spatial Attention with Top-k pooling module for spatial feature extraction. The fused features are further processed using a Graph Convolutional LSTM (GC-LSTM) to capture inter-window dynamics. Evaluated on CHB-MIT and Siena datasets using 10-fold cross-validation, our MFTSGNet model achieved 99.24% accuracy on the CHB-MIT dataset and 99.33% accuracy on the Siena Dataset, outperforming existing methods. For further validation, we test the performance using 10-fold cross-validation and Leave-one-patient-out-cross-validation(LOPOCV) on both CHB-MIT and Siena Datasets, showing high accuracy and stability under various validation schemes. These results demonstrate the model's strong potential for reliable and accurate automated seizure detection in clinical settings.

*Index Terms*—Epileptic seizure detection, EEG analysis, Wavelet Transform, TCN, Graph Learning

Zhiheng Zhang and Xiang Li contributed equally to this work.
*Corresponding author: Shixiong Chen.

This work was supported in part by the National Key RD Program of China (2022YFE0197500), National Natural Science Foundation of China (62471422, 62401558), Shenzhen Medical Research Fund (D2402003), the Science and Technology Innovation Special Fund for Medical Health Technology Research and Development of Longgang District, Shenzhen (LGKCYLWS2024-16), and the Science and Technology Planning Project of Shenzhen(JCYJ20230807093819039).

## I. INTRODUCTION

Epilepsy is a chronic neurological disorder marked by recurrent seizures resulting from abnormal brain activity. Affecting around 50 million people globally, it significantly impairs cognitive function and quality of life, especially in children and the elderly. For patients with drug-resistant epilepsy, timely and accurate seizure detection is critical for effective management and intervention. However, the unpredictable and heterogeneous nature of seizures presents a major challenge for continuous monitoring.

Electroencephalography (EEG), with its high temporal resolution, remains the clinical gold standard for seizure detection. Recent advances in machine learning and deep learning have greatly improved EEG-based methods by enabling automatic learning of abnormal brain activity patterns. Models such as Convolutional Neural Network(CNN) [1], Long Short-term Memory (LSTM) [2], and Temporal Convolutional Network(TCN) [3] have shown strong performance in capturing temporal features. Huang et al. [4] proposed STFFDA, a dual-attention-based model that extracts features directly from raw EEG, improving diagnostic efficiency. Abdulwahhab et al. [5] proposed a deep learning framework that integrates CNN and LSTM networks to detect epileptic seizures using both raw EEG signals and their time-frequency representations generated through Continuous Wavelet Transform(CWT) and Short-Time Fourier Transform(STFT). However, scalp EEG-based seizure detection remains challenging due to the complex, non-stationary nature of EEG signals and the variability across

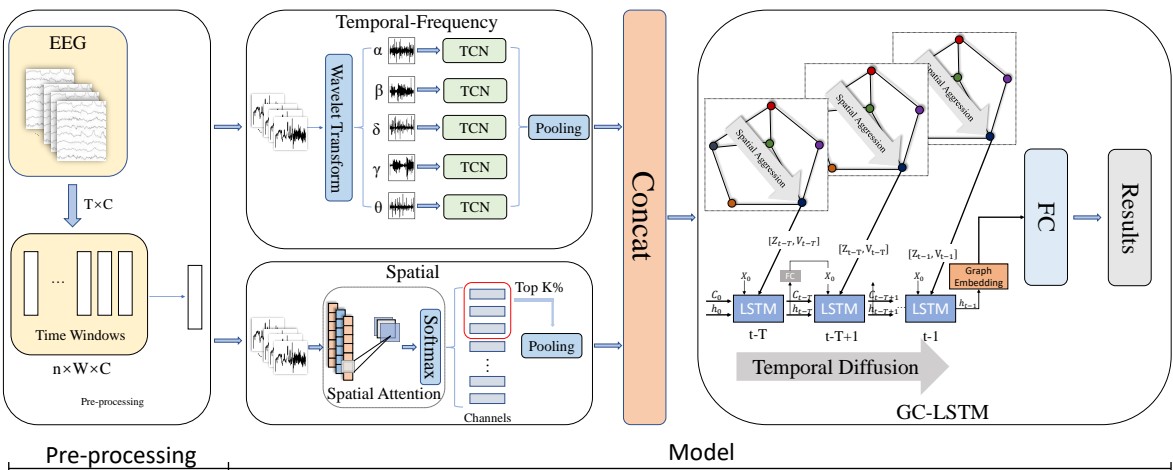

Fig. 1: Overview of the proposed epileptic seizure detection framework. The model consists of a preprocessing stage, a dual-branch feature extractor, and a GC-LSTM classifier.

patients, which complicates the modeling of spatiotemporal dynamics [6]. The spatial topological structure of EEG signals reflects the dynamic coupling characteristics of the brain functional network, while existing methods are difficult to simultaneously model the joint representation of spatial adjacency relationship, temporal dynamic evolution and frequency-domain oscillation features. It is necessary to construct a unified spatio-temporal frequency feature coding module to achieve the collaborative representation of multi-dimensional information.

To address these challenges, we propose a dual-branch framework for robust seizure detection, leveraging both time-frequency and spatial perspectives. Traditionally, detection methods only focus on spatial or temporal features, whereas seizures always contain complex spatial and temporal transformations in EEG signals. As a result, we design a model that takes both spatial and temporal features into account. EEG signals are input into temporal and spatial modules separately. Specifically, our method adopts wavelet transform and a temporal convolutional network to capture frequency-specific dynamics. In parallel, a spatial attention mechanism is applied to select the most informative EEG channels through top-k pooling. The resulting time-frequency and spatial features are concatenated and fed into a Graph Convolutional Long Short-term Memory(GC-LSTM) module, which models both spatial dependencies and temporal transitions across EEG segments. Our model achieves systematic integration of spatiotemporal-frequency characteristics during seizure episodes by modeling the functional connectivity topology among EEG channels and employing time-frequency joint analysis mechanisms, which shows great robustness and performance in seizure detection and promising clinical application prospects.

## II. METHOD

As shown in Fig.1, the model consists of three key components. First, raw EEG signals are segmented into fixed-time windows. A dual-branch module extracts features: the time-frequency branch uses wavelet and temporal convolution for spectral patterns, while the spatial branch employs attention and top-k% to identify critical spatial cues. Combined features are processed by GC-LSTM to model spatiotemporal dynamics for seizure detection.

### A. Time-Frequency Domain

To capture multiscale oscillatory patterns in the EEG signal, each segmented EEG window was processed with Discrete Wavelet Transform (DWT), which decomposed the signal into five sub-bands corresponding to delta, theta, alpha, beta, and gamma frequency components For each decomposed frequency component, we employ an individual TCN to extract high-level temporal features. The formula is as follows:

$$y(t) = \sum_{i=0}^{k-1} w(i) \cdot x(t - d \cdot i) \tag{1}$$

where $k$ is the kernel size, $d$ is the dilation factor, and $w(i)$ denotes the learnable weights of the convolution filter. Separate TCNs learn frequency-specific temporal patterns via stacked dilated causal convolutions, capturing both short- and long-range EEG dynamics. Their outputs are globally averaged to form compact time-frequency features.

### B. Spatial Domain

In parallel with the time-frequency branch, we introduced a spatial attention module to process the segmented windows and learn channel-wise importance weights. Spatial attention

evaluates the relevance of individual EEG channels by summarizing their temporal activation:

$$\mu_c = \frac{1}{T} \sum_{t=1}^{T} X_{c,t}, \quad c = 1, 2, \ldots, C \tag{2}$$

We then apply a 1-D convolution over the channel-mean sequence to generate spatial attention scores:

$$s_c = \sum_{j=-k}^{k} w_j \cdot \mu_{c+j} + b, \quad c = 1, 2, \ldots, C \tag{3}$$

where $s_c$ is the smoothed output; $\mu_{c+j}$ is the input from neighbor channel $c + j$; $w_j$ is the weight; $b$ is the bias; $k$ is the half window size. This approach draws inspiration from SENet [7], which adaptively recalibrates channel-wise features. Unlike SENet's use of fully connected layers, we adopt a lightweight 1-D convolutional layer to capture local channel dependencies, followed by a softmax normalization to produce interpretable attention weights. This convolution-based design achieves similar functionality with fewer parameters and better suits the spatial layout of EEG signals.

Attention scores rank all channels, from which the top-k% are selected. Their signals are pooled and flattened into a spatial feature vector, reducing noise and enhancing efficiency by focusing on discriminative seizure-related patterns.

### C. GC-LSTM network

Inspired by the model of Guo et al. [8], to jointly model the spatial and temporal dynamics of EEG signals, we employ a GC-LSTM architecture, which combines graph convolution and recurrent learning across time. In our framework, each EEG segment is first represented as a graph, where each node corresponds to an EEG channel and edges capture inter-channel correlations based on a predefined or learned adjacency matrix.

At each time step $t$, we apply a Graph Convolutional Network (GCN) to extract spatial features from the input $X_t \in \mathbb{R}^{C \times F}$, where $C$ is the number of EEG channels and $F$ is the feature dimension:

$$H_t = \sigma \left( \tilde{D}^{-\frac{1}{2}} \tilde{A} \tilde{D}^{-\frac{1}{2}} X_t W_g \right) \tag{4}$$

Here, $\tilde{A} = A + I$ includes self-loops, $\tilde{D}$ is the corresponding degree matrix of $\tilde{A}$, and $W_g \in \mathbb{R}^{F \times d}$ is a trainable weight matrix. The output $H_t \in \mathbb{R}^{C \times d}$ captures the spatial relationships among EEG channels at time $t$.

To model the temporal dependencies across EEG segments, the spatial embeddings $\{H_1, H_2, \ldots, H_T\}$ are sequentially passed through a LSTM network:

$$h_t = \text{LSTM}(H_t, h_{t-1}) \tag{5}$$

The final hidden state encapsulates both spatial structure and temporal evolution, and is subsequently used for seizure classification.

TABLE I: Model Architecture Parameters

| Description | Parameters |
|---|---|
| Frequency bands | $\alpha$, $\beta$, $\delta$, $\gamma$, $\theta$ |
| TCN Layers | 5, Kernel: 3×1, Stride: 1 |
| Pooling | Max Pooling, Dimension: $n \times (W/2) \times 64$ |
| LSTM Layer | Units: 128 |

## III. EXPERIMENTS

All experiments were implemented in Python 3.10.12 and PyTorch 2.7.0 on an RTX A6000 (48GB) GPU, using a learning rate of 1e-3 and top-k%=30%. For robustness, we employed 10-fold cross-validation and Leave-One-Patient-Out Cross-Validation (LOPOCV) strategy. The detailed parameter settings for the model are detailed in Table I.

### A. Dataset

CHB-MIT dataset is frequently used in epileptic seizures detection models training, which contains EEG signal records from 24 patients. The EEG signals were recorded using the international 10–20 electrode placement system, with a sampling rate of 256 Hz and up to 23 channels per recording. Among the 24 subjects, 22 (5 males aged 3–22 years and 17 females aged 1.5–19 years) contributed usable seizure data. Each recording includes a mix of seizure and non-seizure periods, allowing for comprehensive evaluation of model performance under real-world conditions.

The Siena Scalp Dataset comprises EEG recordings from 14 patients at the University of Siena's Department of Neurology and Neurophysiology. The cohort includes 9 males aged 25 to 71 and 5 females aged 20 to 58. EEG data were collected using a video-EEG system with a 512 Hz sampling rate and electrodes placed according to the international 10-20 system. Diagnoses of epilepsy and seizure classifications were made by an expert clinician following International League Against Epilepsy standards, based on thorough clinical and electrophysiological data review.

### B. Data Pre-processing

Raw EEG signals were processed by selecting common channels across recordings and applying a 0.5–50 Hz band-pass filter to remove baseline drift and high-frequency noise. ICA was used to eliminate ocular and muscular artifacts. The EEG segments were then normalized with z-score standardization for statistical consistency. Pre-processed data was divided into 2-second overlapping windows with a 1-second stride. To capture temporal dependencies, consecutive windows were grouped into fixed-length sequences for spatiotemporal modeling.

### C. Evaluation

We treat epileptic seizures as the positive class and optimize the decision threshold by maximizing Youden's Index (TPR - FPR) on the validation set. Model performance is evaluated

using accuracy, sensitivity, Kappa, F1 score, and False Positive Rate(FPR) metrics, calculated as follows:

$$\text{Accuracy} = \frac{TP + TN}{TP + TN + FP + FN} \tag{6}$$

$$\text{Sensitivity} = \frac{TP}{TP + FN} \tag{7}$$

$$\kappa = \frac{p_o - p_e}{1 - p_e} \tag{8}$$

$$\text{F1} = 2 \cdot \frac{\text{Adjusted Precision} \cdot \text{Sensitivity}}{\text{Precision} + \text{Sensitivity}} \tag{9}$$

$$\text{FPR} = \frac{FP}{FP + TN} \tag{10}$$

### D. Detailed Validation Strategies

We used two model validation methods: 10-fold cross-validation and leave-one-patient-out-cross-validation (LOPOCV) [10]. In 10-fold-cross-validation, the dataset is divided into 10 equal subsets, with 90% used for training and 10% for testing in each fold. This process repeats 10 times, with each subset serving as the test set once, and performance is averaged across folds for a stable accuracy estimate. For LOPOCV, all EEG segments from one patient are used as the test set, while data from other patients are used for training, simulating real-world clinical scenarios.

## IV. RESULTS

### A. Overall Results

We evaluate our proposed model on the CHB-MIT dataset using the ROC-optimized threshold. The overall performance demonstrates the model's strong capability in detecting epileptic seizures with high reliability and robustness, as presented in Table II. Our model achieved a 99.24% accuracy on epileptic seizure detection using the CHB-MIT dataset with 10-fold validation, demonstrating excellent classification performance. Sensitivity reached 98.15%, indicating strong seizure detection capability. F1 score and Cohen's Kappa were 97.50% and 97.05%, respectively, with an FPR/h of 0.09, showing balanced precision-recall and high agreement with ground truth. In LOPOCV, the model recorded an average accuracy of 90.12% and FPR/h of 0.35, reflecting robust generalization across patients. For the Siena Scalp dataset, 10-fold validation yielded 99.33% accuracy and 0.11 FPR/h, while LOPOCV achieved 91.09% accuracy and 0.42 FPR/h, maintaining competitive performance despite the stricter setting. These results validate the effectiveness and robustness of our framework in handling complex and imbalanced EEG data from both datasets.

### B. Ablation Experiments Results

In order to compare the performance of individual modules with the complete model and to evaluate the effects of top-k on our complete model, we designed the ablation experiments. It contains the ablation experiment of the WT+TCN module and the Spatial Attention module as well as different top-k parameters.

*1) Module Ablation Experiments:* We evaluated the contributions of the temporal (WT+TCN), spatial (Spatial Attention), and complete MFTSGNet modules through ablation studies, as shown in Fig. 2. The WT+TCN module, which captures time-frequency features from EEG signals, achieved 95.23% accuracy, while the Spatial Attention module, responsible for extracting spatial dependencies among EEG channels, achieved 84.25%. When combined, these branches significantly improved classification performance to 99.24%, demonstrating their complementary strengths. The full MFTS-GNet model further integrates both branches with GC-LSTM to capture inter-window temporal dynamics, achieving the best overall performance.

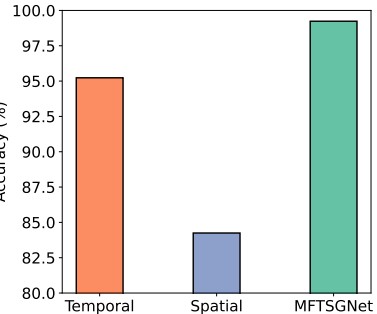

Fig. 2: Model ablation analysis.

*2) Parameter k Ablation Experiment:* To explore the effect of the Top-k% parameter in the Spatial Attention module, we evaluated the model with top-k percentages of 10%, 30%, 50%, 70%, and 90%, corresponding to increasing numbers of selected channels. As shown in Fig. 3, the model performance improves as k increases from 10% to 30%, reaching its peak at 30%. Beyond this point, the performance slightly degrades.

This indicates that selecting the top 30% most informative channels strikes an effective balance between retaining discriminative spatial information and minimizing overfitting or noise amplification. Therefore, we adopt k=30 as the optimal setting in our final model.

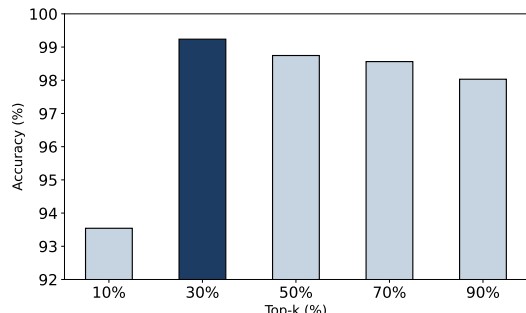

Fig. 3: Top-k% ablation analysis.

TABLE II: Performance of the proposed model under different validation settings on the CHB-MIT and Siena Scalp datasets

| Dataset | Method | Accuracy (%) | Sensitivity (%) | F1 Score (%) | Kappa (%) | FPR/h |
|---|---|---|---|---|---|---|
| CHB-MIT | 10-Fold | 99.24 | 98.15 | 97.50 | 97.05 | 0.09 |
| | LOPOCV | 90.12 | 90.64 | 94.53 | 93.33 | 0.35 |
| Siena Scalp | 10-Fold | 99.33 | 99.30 | 98.17 | 98.88 | 0.11 |
| | LOPOCV | 91.09 | 91.23 | 92.46 | 91.93 | 0.42 |

TABLE III: Performance comparison with existing methods

| Authors | Method | Year | Accuracy (%) | Sensitivity (%) | F1-Score (%) | Kappa (%) |
|---|---|---|---|---|---|---|
| Ma et al. [3] | MCFF-CNN-Bi-LSTM | 2023 | 94.83 | 94.84 | 94.83 | - |
| Zhu et al. [11] | SE-TCN-BiGRU Hybrid Network | 2024 | 98.77 | 95.88 | 96.77 | - |
| Zhang et al. [12] | DLFE | 2024 | 98.63 | 98.06 | **99.00** | 97.00 |
| Chen et al. [13] | Spiking Conformer | 2024 | 97.10 | 94.90 | - | - |
| Diao et al. [14] | TFANet | 2025 | 97.00 | 94.51 | 96.99 | 93.19 |
| **MFTSGNet** | MFTSGNet | **2025** | **99.24** | **98.15** | 97.50 | **97.05** |

## C. Analysis of Spatial Attention for Clinical Interpretability

To assess the clinical interpretability of MFTSGNet, we analyzed the effectiveness of the Spatial Attention module by visualizing its learned channel weights during seizure episodes. As illustrated in the topographic maps in Fig. 4, the model demonstrates a clear ability to differentiate the contribution of individual EEG channels with high spatial specificity. The weighting scheme reveals that the model dynamically adapts to focus on distinct, clinically relevant scalp regions that correlate with seizure activity for each patient. Notably, the spatial distribution of high-attention areas varies across individuals, demonstrating the model's capability to capture patient-specific ictal patterns rather than applying a uniform, generalized weighting. This focal attention mechanism closely mirrors clinical diagnostic reasoning, where neurologists localize epileptogenic zones based on individualized EEG manifestations. The model's ability to selectively emphasize different brain regions across patients enhances its clinical relevance, as it effectively emulates the expert process of identifying personalized epileptic foci.

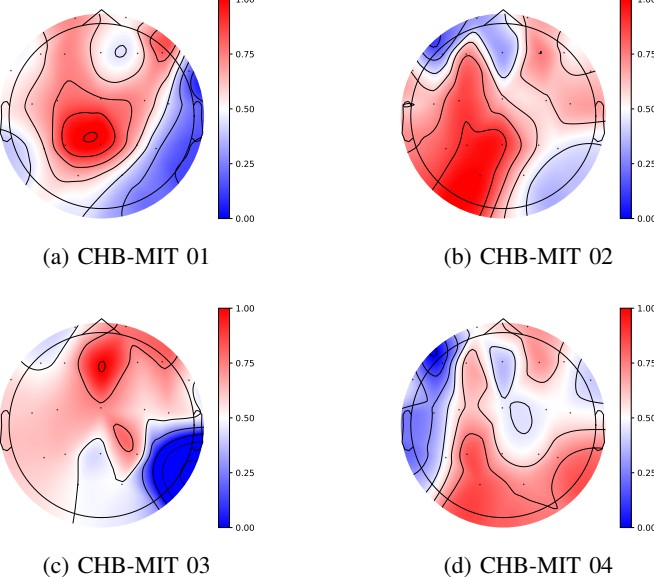

(a) CHB-MIT 01      (b) CHB-MIT 02

(c) CHB-MIT 03      (d) CHB-MIT 04

Fig. 4: Attention weights for CHB 01-04 during seizure

## D. Comparison With Other Advanced Methods

We evaluated the performance of our proposed MFTSGNet model on the CHB-MIT dataset and compared it with several advanced methods to demonstrate its effectiveness and robustness, as shown in Table III. Compared with the TFANet model proposed by Diao et al. [11], our model achieves consistent improvements across all evaluation metrics, with the most notable gains being 2.24% in Accuracy and 3.66% in Sensitivity. These results highlight the superior performance and strong generalization ability of our model. The consistent gains over a advanced baseline demonstrate that MFTSGNet is both effective and reliable for real-world seizure detection tasks.

## V. CONCLUSION

In this paper, we developed MFTSGNet, a novel epileptic seizure detection framework that integrates both time-frequency and spatial information, and leverages graph-based temporal modeling through a GC-LSTM architecture. The proposed model is designed to fully exploit the multi-dimensional characteristics of EEG signals by combining wavelet-based time-frequency analysis, spatial attention mechanisms, and graph-based sequential learning. Extensive experiments on the CHB-MIT dataset demonstrate that our model achieves superior performance in terms of accuracy, sensitivity, F1-score, and Kappa coefficient compared to existing state-of-the-art methods.

Future work will explore generalization to clinical datasets. Moreover, we plan to evaluate MFTSGNet on mobile EEG platforms (e.g., OpenBCI) with real-time streaming pipelines, and assess its robustness under clinical noise conditions, such as motion artifacts and hardware variability. Additionally, incorporating adaptive graph structures and multimodal data may further enhance its accuracy and resilience.

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
