# OpenReview forum: "MFTSGNet: A Dual-Branch Spatio-Temporal Network for Robust Seizure Detection"
_IEEE.org/EMBS/BHI/2025/Conference — BHI 2025_

### Official Review · Reviewer_Ad4t · 2025-07-17
**Review of MFTSGNet: A Dual-Branch Spatio-Temporal Network for Robust Seizure Detection**

**Confidence:** 3
**Clarity Of Writing:** fair
**Clinical Significance:** good
**Methodological Novelty:** good
**Overall Rating:** 4
**Final Rating:** 5

**Experiments And Results:**

fair

**Questions For The Authors:**

Did you split the folds by patients or by recording? If windows from the same seizure appear in both the train and test sets, performance may be overestimated. A patient-wise split could materially lower metrics, which could affect the experiments and results.

**Strengths:**

Combining wavelet based TCNs with adaptive spatial attention addresses EEG’s inherent spectral and spatial heterogeneity. The design is holistic and appropriate.

GC LSTM architecture integrates adjacency structured spatial features with sequential context, a meaningful advance over LSTM pipelines.

The extensive ablations (module removal, k% sweep) and comparison against six recent methods clearly demonstrate incremental gains.
Detecting 98 % of seizure windows is clinically valuable, reducing missed events and shows high sensitivity.

**Summary Of The Paper:**

The authors introduce MFTSGNet, a seizure detection framework that processes scalp EEG through two complementary branches:  time-frequency and spatial.

**Weaknesses:**

The unexplained abbreviation in the title makes it hard to read and understand the title quickly: Multi-Frequency TCN Spatial Graph Net-work (MFTSGNet)

The authors could verify robustness under stricter patient-wise splits, add false alarm metrics, and validate on additional datasets or higher density montages.

They used a single pediatric dataset (CHB-MIT dataset is a dataset of EEG recordings from pediatric subjects). Age, skull thickness, electrode layout, seizure semiology, and hardware noise differ significantly between pediatric and subjects. So the developed model may learn specific channel noise patterns or seizure morphologies that do not generalize to adults, dense montages, or other institutions. Decreases clinical significance and deployment potential. Clinical deployment could fail when encountering unseen demographics. The authors could try to fine-tune on a small adult cohort and measure performance to show transferability

There is a typo in the conclusion's first line “epiletic"

---

### Official Review · Reviewer_jK1j · 2025-07-18
**MFTSGNet: A Dual-Branch Spatio-Temporal Network for Robust Seizure Detection**

**Confidence:** 5
**Clarity Of Writing:** fair
**Clinical Significance:** good
**Methodological Novelty:** good
**Overall Rating:** 4
**Final Rating:** 6

**Experiments And Results:**

fair

**Questions For The Authors:**

How does the model perform across different seizure types in the CHB-MIT dataset?

Can you provide a detailed architectural diagram or table showing the full configuration of each module?

What are the specific limitations in current state-of-the-art models that your work addresses, and how does your approach overcome them?

**Strengths:**

Innovative Dual-Branch Architecture:
The proposed MFTSGNet introduces a well-structured dual-branch framework that separately captures temporal-frequency features using WT+TCN and spatial dependencies using attention with top-k pooling. This modular design effectively addresses the non-stationary and complex nature of EEG signals in seizure detection.

Incorporation of Graph-Based Sequential Modeling:
The integration of a Graph Convolutional LSTM (GC-LSTM) allows the model to learn spatial relationships among EEG channels and temporal dynamics across segments, offering a powerful and flexible method for modeling EEG spatio-temporal patterns.

**Summary Of The Paper:**

This paper introduces MFTSGNet, a novel dual-branch spatio-temporal neural network designed for robust epileptic seizure detection from EEG signals. The model integrates a time-frequency branch (combining Discrete Wavelet Transform and Temporal Convolutional Networks) and a spatial branch (featuring spatial attention and top-k channel pooling). The extracted features are fused and passed through a GC-LSTM module, capturing both inter-channel spatial relationships and temporal evolution across EEG windows.

**Weaknesses:**

No Subject-Wise Split for Validation:
The use of 10-fold cross-validation is reasonable, but the paper does not clarify whether the data split is subject-independent. Since EEG signals can be highly subject-specific, this is crucial for validating the model’s real-world usability across unseen patients. If not subject-independent, the reported performance may be overly optimistic.

No Discussion of Seizure Type Variability:
The CHB-MIT dataset includes different types of seizures, but the paper does not analyze performance by seizure subtype. Understanding whether the model generalizes across various seizure morphologies would add significant value.

Lack of Detailed Architectural Description
The paper provides a high-level overview of the model architecture, particularly for the attention-based fusion design, but does not include a complete diagram or detailed layer-by-layer breakdown. This omission limits reproducibility and makes it difficult to fully understand or replicate the implementation.

No Discussion of importance of this work and the state-of-the-art limitations.

Clarity and Writing Issues
While the overall structure is strong, the paper contains minor typos, occasional awkward phrasing, and some inconsistencies (e.g., referring to modules before defining them). This detracts slightly from readability and polish.

---

### Official Review · Reviewer_kncZ · 2025-07-20
**The paper proposes a robust EEG-based seizure detection model combining multi-resolution time-frequency analysis and spatial learning. MFTSGNet leverages domain-specific signal decomposition (Wavelet), deep temporal modeling (TCN), attention-based spatial feature extraction, and GC-LSTM for sequential dynamics. Achieves state-of-the-art performance.**

**Confidence:** 5
**Clarity Of Writing:** great
**Clinical Significance:** great
**Methodological Novelty:** excellent
**Overall Rating:** 7

**Experiments And Results:**

excellent

**Questions For The Authors:**

* Can you provide interpretability for the attention weights—what EEG regions does the model consider most important?
* How do you ensure robustness across recording variations or patient types (e.g., age/gender)?
* Have you tested the system on real-time streaming EEG or mobile EEG devices?
* What challenges would you face in deploying this in a clinical setting with noisy data?
* How sensitive is the model to different artifact removal strategies (e.g., ICA variants)?

**Strengths:**

*  **High clinical relevance**: Focuses on epilepsy, a major neurological condition.
* **Novel architecture**: Combines wavelet, TCN, spatial attention, and graph-based LSTM – rarely integrated together.
*  **Excellent accuracy & robustness**: State-of-the-art performance on a benchmark dataset.
* **Comprehensive ablation study**: Justifies module contributions and top-k tuning.
* **Practical preprocessing pipeline**: Uses ICA, band-pass filtering, and z-score normalization—industry-aligned techniques.

**Summary Of The Paper:**

The paper proposes a robust EEG-based seizure detection model combining multi-resolution time-frequency analysis and spatial learning. MFTSGNet leverages domain-specific signal decomposition (Wavelet), deep temporal modeling (TCN), attention-based spatial feature extraction, and GC-LSTM for sequential dynamics. Achieves state-of-the-art performance.
MFTSGNet processes EEG signals using two parallel modules:

* Time-Frequency Branch: Discrete wavelet transform (DWT) splits signals into delta–gamma bands. Each is processed by a separate **TCN** to capture frequency-specific dynamics.
* Spatial Branch: A lightweight **spatial attention mechanism with top-k pooling** selects significant EEG channels based on importance.
* Combined features are passed through **GC-LSTM**, a graph-based LSTM model modeling spatial and temporal dependencies.

Evaluated on the CHB-MIT EEG dataset with 10-fold CV, the model achieved:

* Accuracy: 99.24%, Sensitivity: 98.15%
* F1 Score: 97.50%, Cohen’s Kappa: 97.05%

**Weaknesses:**

* **Lacks interpretability**: No visualization or clinical explanation of attention weights or decision-making.
* **No real-time testing**: Doesn't discuss inference speed or deployment feasibility in real-world EEG monitoring systems.
* **Generalization not demonstrated**: Only CHB-MIT dataset used; results may not transfer to other populations or recording systems.
* **No physician involvement or clinical annotation feedback.**

---

### Official Review · Reviewer_MBUm · 2025-07-20
**Review of MFTSGNet**

**Confidence:** 4
**Clarity Of Writing:** great
**Clinical Significance:** fair
**Methodological Novelty:** good
**Overall Rating:** 7

**Experiments And Results:**

great

**Questions For The Authors:**

The performance of the proposed method is better than previous method. However, the improvement is not very obvious in metrics. Does the improvement have significant in clinical medicine?

**Strengths:**

The paper is well-structured and with clear writing. The innovative combination of GNNs and LSTM in the proposed method shows great promise.

**Summary Of The Paper:**

This paper proposed a novel spatio-temporal network for seizure detection and achieved better performance than previous study.

**Weaknesses:**

The arrangement of elements and text size in Figure 1 can be further optimized